# `LightVeriFL`: Lightweight and Verifiable Secure Federated Learning

**Baturalp Buyukates**
University of Southern California
buyukate@usc.edu

**Jinhyun So**
University of Southern California
jinhyuns@usc.edu

**Hessam Mahdavifar**
University of Michigan
hessam@umich.edu

**Salman Avestimehr**
University of Southern California
avestime@usc.edu

## Abstract

Secure aggregation protocols are implemented in federated learning to protect the local models of the participating users so that the server does not obtain any information beyond the aggregate model at each iteration. However, existing secure aggregation schemes fail to protect the integrity, i.e., correctness, of the aggregate model in the possible presence of a malicious server forging the aggregation result, which motivates the need for verifiable aggregation in federated learning. Existing verifiable aggregation schemes either have a complexity that linearly grows with the model size or require time-consuming reconstruction at the server, that is quadratic in the number of users, in case of likely user dropouts. To overcome these limitations, we propose `LightVeriFL`, a lightweight and communication-efficient secure verifiable aggregation protocol, that provides the same guarantees for verifiability against a malicious server, data privacy, and dropout-resilience as the state-of-the-art protocols without incurring substantial communication and computation overheads. The proposed `LightVeriFL` protocol utilizes homomorphic hash and commitment functions of constant length, that are independent of the model size, to enable verification at the users. In case of dropouts, `LightVeriFL` uses a one-shot aggregate hash recovery of the dropped users, instead of a one-by-one recovery based on secret sharing, making the verification process significantly faster than the existing approaches. We evaluate `LightVeriFL` through experiments and show that it significantly lowers the total verification time in practical settings.

## 1 Introduction

Federated learning (FL) is a distributed learning paradigm proposed to address the growing concerns about user data privacy in distributed learning systems [1]. In FL, a group of users jointly train a global model without sending their local data to a central server (see Fig. 1(a)). Even though user datasets stay private, local models sent by the users can potentially cause data leakage from the users [2, 3, 4, 5]. Secure aggregation frameworks have been implemented to protect the users' individual local models as well as tolerate likely dropouts in FL [6, 7, 8, 9, 10]. Secure aggregation schemes hide individual local models from the server, which only learns the aggregate model. Despite their benefits in protecting the user models, none of these secure aggregation schemes enable the users to verify the correctness of the aggregate model received from the server at each iteration.

The typical FL framework, even in the presence of secure aggregation schemes, is prone to a malicious server forging the aggregation results for its own benefit or a lazy server sending incorrect results

Workshop on Federated Learning: Recent Advances and New Challenges, in Conjunction with NeurIPS 2022 (FL-NeurIPS'22). This workshop does not have official proceedings and this paper is non-archival.

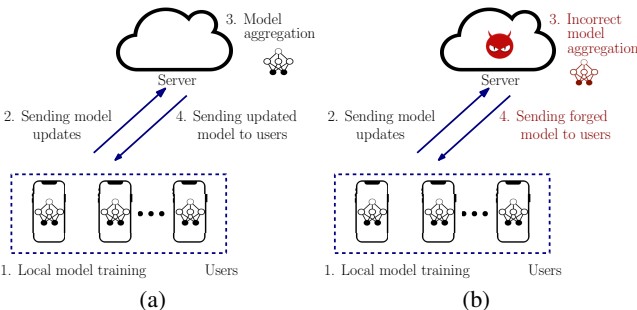

Figure 1: (a) Typical federated learning framework, (b) a malicious server can forge the aggregation results as the users cannot verify the integrity of the incoming aggregation result.

to reduce its computation cost (see Fig. 1(b)). Since the users are oblivious to the aggregation procedure, such incorrect computations at the server can potentially alter the learning procedure. From a trustworthiness standpoint, without verifiable aggregation, users cannot make sure whether their contributions are included in the global model, also motivating the study of verifiability in FL.

**Related Works.** In [11], the server generates a proof for the aggregation computation to enable verification at the users. The proof utilized in [11] has a communication overhead that is linear in the model size, which makes it impractical in modern FL systems with hundreds of thousands of parameters. Authors in [12] propose a communication-efficient verifiable aggregation scheme, which utilizes hashes of the local model updates of the users for verification at the expense of huge computation overhead in the presence of dropouts (see Appendix A for further discussion). In this work, we propose a lightweight verifiable aggregation scheme that provides the same guarantees for input privacy, dropout resilience and verifiability in the presence of a malicious server as the state-of-the-art protocols without incurring substantial computation and communication overheads.

**Contributions.** The main bottleneck in [12] is that the server recovers the hashes of the dropped users one-by-one in the verification stage (see Appendix C for details). These hashes are later utilized by the users to check the integrity of the aggregate model. In large systems with frequent dropouts, this one-by-one reconstruction incurs a significant quadratic burden on the verification time. In this work, inspired by the design of `LightSecAgg` [8], we propose a verifiable aggregation scheme named `LightVeriFL` for faster verification in the presence of dropouts in FL systems.

The proposed `LightVeriFL` scheme utilizes linearly homomorphic hashes of the local models of the users for verification. As shown in Fig. 2, after the local training, each user generates a hash, which is protected by a random mask generated by the respective user. These masks are encoded carefully such that once received sufficient responses from the users, the server is able to generate the aggregate hash of all participating users in one-shot (as opposed to one-by-one reconstruction of [12]). That is, even if certain number of users drop in the verification stage, the server is able to recover the aggregate hash of all users that have sent model updates in that iteration. Finally, enabled by the linear homomorphism of the hashes, each surviving user verifies the integrity of the aggregation. Key features of the proposed `LightVeriFL` protocol are listed as follows:

**(1) Verifiability and trustworthiness.** With `LightVeriFL` users make sure that i) their contributions are reflected in the global model and ii) the aggregation result provided by the server is exact.

**(2) Compatibility with the existing secure aggregation schemes.** `LightVeriFL` is compatible with federated averaging-based secure aggregation schemes, e.g., `SecAgg` [6], `LightSecAgg` [8].

**(3) Input privacy and dropout resilience.** `LightVeriFL` does not leak any private information of the users and is resilient to dropouts during the verification procedure. Combined with a secure aggregation scheme, `LightVeriFL` forms a secure verifiable aggregation scheme and guarantees input privacy and dropout resilience both in model aggregation and aggregate model verification.

**(4) One-shot aggregate hash recovery at the server.** In `LightVeriFL`, the server is able to recover the aggregate hash of the participating users (whose models are included in the aggregation) in one-shot even in the presence of dropouts thanks to the employed mask encoding strategy.

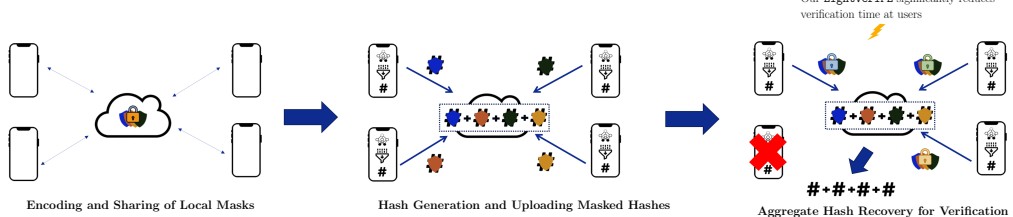

Figure 2: Illustration of the proposed `LightVeriFL` protocol. (1) Users first encode and share their local masks among themselves. (2) Each user generates hash of its local model and uploads the masked hash to the server. (3) In the verification stage, the surviving users upload the aggregate encoded masks to the server, which then recovers the desired aggregate mask. By cancelling out the aggregate mask, the server recovers the aggregate hash, which is used by the users to verify the integrity of the aggregation.

**(5) Reduced overheads and faster verification.** With the one-shot aggregate hash recovery, computation overhead at the server drastically decreases in case of dropouts compared to [12], which results in much faster verification, also confirmed by our empirical results for realistic model sizes, particularly in systems with large number of users. Our experiments indicate that the primary source of this gain is the complexity reduction at the server.

**(6) A novel encoding scheme.** Existing encoding strategies for secure aggregation aim at recovering the sum of the individual models. In the case of linear homomorphic hashes, one needs the product of the hashes of the users for verification. Inspired by the encoding strategy in [8], we propose a novel encoding strategy that utilizes elliptic curves to recover the "aggregate" product of the user hashes.

**Notation.** $\mathbb{Z}_p$ denotes the ring of integers modulo $p$. We use $\mathbb{Z}_p^*$ to denote all invertible elements of $\mathbb{Z}_p$, i.e., $\mathbb{Z}_p^* = \mathbb{Z}_p \backslash \{0\}$. $\mathbb{G}$ denotes a cyclic group of order $q$. We have $[N] \triangleq \{1, 2, \dots, N\}$.

## 2 Preliminaries

**Linear homomorphic hash.** Following [12, 13], we let $\mathbb{G}$ be a cyclic group with prime order $q$ and generator $g$. Given $d$ distinct elements $g_1, g_2, \dots, g_d$, the hash of a model $\mathbf{x} \in \mathbb{F}_q^d$ is given by

$$h(\mathbf{x}) \leftarrow \prod_{j=1}^d g_j^{\mathbf{x}[j]} \in \mathbb{G}, \tag{1}$$

where $\mathbf{x}[j]$ denotes the $j$th element of the model vector $\mathbf{x}$. The hash construction in (1) has collision resistance [13] such that it is computationally infeasible to find two distinct vectors $\mathbf{x_1}, \mathbf{x_2} \in \mathbb{F}_q^d$ that satisfy $h(\mathbf{x_1}) = h(\mathbf{x_2})$. The resulting hash in (1) is of constant length independent of $d$ and satisfies an additivity (in exponent) property for $\mathbf{x_1}, \mathbf{x_2} \in \mathbb{F}_q^d$ such that $h(\mathbf{x_1} + \mathbf{x_2}) = h(\mathbf{x_1})h(\mathbf{x_2})$.

**Commitment.** A commitment scheme COM, takes an input message $h$ and uniform randomness $r$ to produce a commitment string $c$ such that $c = \text{COM.Commit}(h, r)$. When it is time to decommit, i.e., reveal the hidden committed value, the committer sends the claimed committed message $h'$ and the claimed committed randomness $r'$ to the interested party, which then checks $c = \text{COM.Commit}(h', r')$. If this holds, the interested party accepts the committed value $h$.

A well-known commitment scheme is the Pedersen commitment scheme [14]. Given a subgroup $\mathbb{G}$ of $\mathbb{Z}_p^*$ of order $q$, with $p = 2q + 1$, in the Pedersen commitment scheme, the committer sends $c = g^h t^r$, where $g$ is the generator of the subgroup and $t$ is selected such that $t = g^a$ with $a$ unknown to the receiver. Here, $r \in \mathbb{Z}_p^*$ is randomly selected and called the blinding factor. Pedersen commitment schemes are perfectly hiding and computationally binding [14] and are additively homomorphic such that for commitment pairs $(h_1, r_1)$ and $(h_2, r_2)$, we have $c(h_1 + h_2, r_1 + r_2) = c(h_1, r_1)c(h_2, r_2)$.

In `LightVeriFL`, we use a variation of the original Pedersen commitment scheme (see Appendix D) and utilize its homomorphic property to verify the integrity of the hashes reconstructed by the server.

**Elliptic curve (EC).** An EC over $\mathbb{F}_p$, denoted by $E(\mathbb{F}_p)$, consists of points $P = (x, y)$, $x, y \in \mathbb{F}_p$ that satisfy $y^2 = x^3 + ax + b$, together with the point at infinity $\mathcal{O}$ [15]. $p > 3$ is an odd prime and

$a, b \in \mathbb{F}_p$ satisfy $4a^3 + 27b^2 \neq 0$. Two operations are defined on ECs: point addition and scalar multiplication. Given an integer $k$, the scalar multiplication $kP$ corresponds to adding point $P$ to itself $k$ times and is analogous to the exponentiation operation in multiplicative groups. ECs are well-suited for resource constrained environments such as FL since significantly smaller parameters are required to achieve the same level of security, compared to the classical public-key schemes [15].

# 3    Problem Setting

## 3.1    Federated Learning

FL is a distributed learning framework, in which a global model $\mathbf{x}$ of dimension $d$ is jointly trained by a group of users on their own privately held datasets $\mathcal{D}_i$. The FL framework aims to minimize the global loss function $L(\mathbf{x}) = \frac{1}{N}\sum_{i=1}^{N} L_i(\mathbf{x})$, where $L_i(\mathbf{x})$ denotes the local loss function of the $i$th user and without loss of generality, $|\mathcal{D}_i| = n$ for all $i \in [N]$. Training in FL is an iterative process. At each iteration, the server sends out the current global model $\mathbf{x}(t)$ to the participating users. Each user $i$ updates its local model $\mathbf{x_i}(t)$ and sends it to the server. We let $\mathcal{U}_a(t)$ denote the surviving users at iteration $t$ during aggregation. The server aggregates the results with $\mathbf{x}(t+1) = \frac{1}{|\mathcal{U}_a(t)|}\sum_{i \in \mathcal{U}_a(t)} \mathbf{x}_i(t)$ and pushes the updated global model, $\mathbf{x}(t+1)$ back to the users for the next iteration.

## 3.2    Threat Model and Privacy & Verifiability Guarantees

All users and the server are honest but curious. Up to $T$ of the users can collude with each other as well as the server to obtain information on the inputs of the honest users. Corrupted parties follow the protocol and report their local models honestly but they may try to infer the local models of the honest users. Additionally, we allow a corrupted server to forge the aggregation results arbitrarily in an effort to convince the users of a wrong aggregation result. Thus, the goal is to protect the confidentiality of the user inputs as well as give each user the capability of verifying the integrity of the server aggregation.

## 3.3    Dropout Resilience

In FL, users may sometimes drop from the protocol execution due to communication/connection issues, battery problems etc. The proposed secure verifiable aggregation protocol should be resilient to these random dropouts. We assume that at most $D$ users drop during the verification protocol such that we have at least $N - D$ surviving users that want to verify the aggregation. Since the existing secure aggregation schemes [6, 7, 8] provide resilience for dropouts occurring during model aggregation, in this work, we focus on the dropouts occurring during the verification of the aggregate model. Thus, we want the proposed protocol to tolerate $D$ dropouts such that the remaining $N - D$ clients can correctly verify the integrity of the aggregation that includes local models of $N$ users.

**Goal.** We want to design a lightweight and communication-efficient verifiable aggregation protocol that simultaneously provides input privacy against $T = \frac{N}{2}$ colluding users and resilience to $D = \frac{N}{2}-1$ dropouts as well as a verifiability guarantee in the presence of a malicious server spoofing the aggregation results. The proposed verifiable aggregation protocol should be compatible with the existing secure aggregation protocols to protect the confidentiality of the users' local models.

# 4    Overview of the `LightVeriFL` Protocol

Our protocol utilizes certain cryptographic primitives as in [6, 8, 11, 12] so that all the operations are performed over a finite field. In order to implement `LightVeriFL`, users perform the following additional operations during an FL iteration (for the detailed description, complexity analysis, and the pseudo code of `LightVeriFL` see Appendices D, G, and E, respectively).[1]

**During the aggregation phase.** Each user $i$ i) generates a mask $z_i$, encodes it according to (8) and shares the encoded mask with the other users, ii) generates its hash $h_i$ based on its local model $\mathbf{x}_i$ according to (1), iii) commits its hash $h_i$ and exchanges its commitment $c_i$ with the other users, iv)

---

[1]As in `VeriFL`, we implement `LightVeriFL` together with `SecAgg` [6] by default according to Appendix B.

sends its masked hash $\tilde{h}_i = h_i + z_i$ to the server. At the end of the aggregation phase, the server recovers the aggregate model $\mathbf{y}$ and sends it back to the users.

**After the aggregation phase.** Upon the aggregation phase, some users may drop and the verification is performed by the surviving users. Each of the surviving users sends the encoded masks it has received from the other users (surviving and dropped) to the server for reconstruction. Server reconstructs the aggregate mask and recovers the aggregate hashes of the surviving and dropped users (decommitting round). Then, each surviving user i) verifies the correctness of the recovered hashes coming from the server using the commitments it has received from every other user during the aggregation phase, ii) verifies the integrity of the aggregate model $\mathbf{y}$ by computing its hash and comparing it against the aggregate of the individual hashes of the users, reconstructed by the server and accepted by the user in step i) (batch checking round). If a user encounters an incorrect result either in step i) or ii) during the verification phase, it regards the result as forged and rejects the aggregation result $\mathbf{y}$ computed by the server in that iteration.

**Improved `LightVeriFL` with amortized verification:** In `LightVeriFL`, the most time-consuming operation is the two hash computations over a model of dimension $d$. First, in the aggregation phase, each user computes the hash of its local model. Then, in the batch checking round, each user computes the hash of the aggregate model. The former hash computation is necessary, however, the latter one can be amortized in order to cut down the computational overhead of the entire scheme. For this, we implement the amortized verification technique [12] with batch size $L$ such that users verify aggregations of the past $L$ iterations all at once by performing a single hash computation using (13).

## 5 Theoretical Guarantees

In this section, we show the correctness of the verification along with input privacy guarantee and dropout resilience in `LightVeriFL`. Proofs are deferred to Appendix F.

**Theorem 1.** *[Correctness of Verification] Under the `LightVeriFL` scheme, users accept the aggregation results $\mathbf{y}(\ell)$, $\ell \in [L]$ if and only if these results are correctly aggregated by the server with probability (almost) 1.*

**Theorem 2.** *[Input Privacy Guarantee] The proposed `LightVeriFL` protocol provides input privacy against up to $T$ colluding users.*

**Theorem 3.** *[Dropout Resilience in Verification] The proposed `LightVeriFL` scheme guarantees dropout resilience up to any $D$ dropped users during the verification phase such that $N \geq T + D + 1$.*

**Theorem 4.** *The proposed `LightVeriFL` scheme guarantees successful aggregation integrity verification in the presence of any $D$ user dropouts during the verification phase without sacrificing input privacy against up to any $T$ colluding users for $T + D < N$. When `LightVeriFL` is implemented together with a secure aggregation scheme, a secure verifiable aggregation scheme is obtained.*

## 6 Experimental Results

### 6.1 Experimental Setup

**Implementation.** We implement the linearly homomorphic hash as well as the commitments using the NIST P-256 elliptic curve [16]. This curve has a 256-bit subgroup order $n$. We fix $q$ to $2^{31} - 1$ such that all user models lie in $\mathbb{F}_q^d$. As in [12], we simulate the clients and the server on our in-house 64-bit Ubuntu 20.04.2 LTS machine equipped with AMD EPYC 7502 CPU.

**Baseline.** We use the `VeriFL` [12] scheme described in Appendix C as our baseline. We do not consider other aggregation verification methods such as [11] and [17] as baselines since these works are not communication-efficient, i.e., the required communication scales with the model size $d$, and/or do not support user dropouts. In the experiments, we implement both `VeriFL` and `LightVeriFL` on top of `SecAgg` and utilize amortized verification in both schemes with $L = 10$.

**Number of users and dropout rate.** We have up to $N = 200$ users in our experiments. For the dropped users, we consider a worst-case scenario for verification, in which we assume all $N$ users successfully participated in the model aggregation and some $pN$ portion of these users drop in the verification phase. For this, we artificially drop $pN$ users at each iteration. Following the observations made in [8] and [18], we take $p = 0.1$, $p = 0.3$, and $p = 0.5$. In all these cases, we take $T = \frac{N}{2}$.

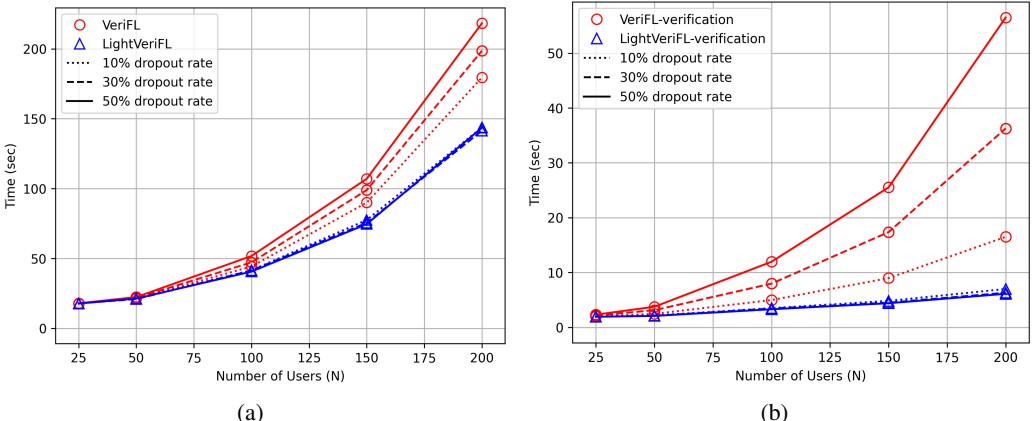

(a)                             (b)

Figure 3: (a) Total running time and (b) the verification phase time comparison of `VeriFL` and the proposed `LightVeriFL` for varying $N$ and dropout rates for $L = 10$. Both schemes are implemented on top of `SecAgg`. `LightVeriFL` specifically targets improving the verification time as it enables one-shot recovery of the user hashes as opposed to one-by-one recovery in `VeriFL`.

Table 1: Breakdowns of the verification time of `LightVeriFL` and `VeriFL` for varying dropout rates with $N = 200$ users, $d = 100K$, and $L = 10$. All times are in seconds.

|  | **Phase** | **10% dropout** | **30% dropout** | **50% dropout** |
|---|---|---|---|---|
| `VeriFL` | Round V.0 Decommitting | 11.25 | 30.56 | 52.51 |
|  | Round V.1 Batch Checking | 6.53 | 6.03 | 5.77 |
|  | Verification Phase - Total | $17.78 \pm 0.14$ | $36.59 \pm 0.25$ | $58.28 \pm 0.60$ |
| `LightVeriFL` | Round V.0 Decommitting | 0.76 | 0.67 | 0.67 |
|  | Round V.1 Batch Checking | 6.46 | 5.94 | 5.54 |
|  | Verification Phase - Total | $7.22 \pm 0.07$ | $6.61 \pm 0.06$ | $6.21 \pm 0.02$ |
| Gain |  | $2.46\times$ | $5.54\times$ | $9.38\times$ |

**Model size.** By default, we use $d = 100K$ as our model size in the experiments as in [12] (see Appendix H for experiment results with varying $d$).

## 6.2 Performance Analysis

We consider a single training round and measure the total running time of `LightVeriFL` and `VeriFL`. We do not include model training in the results shown in this section.

In Fig. 3 we increase the number of users to $N = 200$ for varying dropout rates with $d = 100K$ and $L = 10$. In Fig. 3(a), we see that, unlike `LightVeriFL`, the total running time of the `VeriFL` scheme is affected by larger dropouts as its reconstruction complexity increases quadratically with $N$. To better observe this, in Fig. 3(b), we present the running times of the verification phases of the two schemes. Verification phase in both schemes involves the decommitting and batch checking. In Fig. 3(b), we observe that while the verification time in the proposed `LightVeriFL` scheme is largely unaffected by the increasing dropout rates, the verification time in `VeriFL` significantly increases as $N$ gets larger and as the dropout rate increases. This is due to the fact that `VeriFL` performs one-by-one reconstruction of the dropped user hashes whereas in `LightVeriFL` the server reconstructs the aggregate hash of all users all at once, independent of the dropout rate.

Next, we present the verification time breakdown of the two schemes for different dropout rates with $N = 200$, $d = 100K$, and $L = 10$ in Table 1 as `LightVeriFL` specifically targets improving the verification time. In Table 1, we observe that the proposed `LightVeriFL` protocol achieves up to $9.38\times$ improvement in the verification time (see Appendix H for further discussion).

## 7 Conclusion and Future Directions

Unlike the existing verifiable aggregation schemes which suffer from large communication and computation overheads, the proposed `LightVeriFL` scheme is lightweight and communication-

efficient, which are enabled by the use of constant-length hashes for aggregation verification and one-shot aggregate hash recovery at the server instead of one-by-one recovery of the dropped user hashes. `LightVeriFL` achieves significantly faster aggregate model integrity verification at the users in the presence of a malicious server forging the aggregation results while guaranteeing the same input privacy and dropout-resilience as the state-of-the-art protocols. Despite its benefits, limitations still exist. We plan to improve `LightVeriFL` in the following aspects: 1) performing end-to-end experiments by also considering model training to investigate the gain achieved by `LightVeriFL` over baselines, 2) extending `LightVeriFL` to asynchronous FL, 3) considering the verification of user data and models as well as design of a Byzantine-robust secure verifiable aggregation scheme.

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

# Appendix

## A  Other Related Works

Besides [11] and [12], there are other works in verifiable federated learning focusing on verifying the computations of the server and/or the local information and model updates of the users (see the survey in [19]). Among these, a related work is [17], which uses Lagrange interpolation and the Chinese Remainder Theorem to encrypt the model updates of the users. The server aggregates the encrypted models, which are then verified by the users. A major disadvantage of this scheme is that it cannot support user dropouts, which is one of our main considerations in this work. The aggregate model verification problem that we consider in this work can be cast as a secure multiparty computation in the presence of malicious parties. The existing techniques in the secure multiparty computation domain utilize garbled circuit-based approaches through Cut-and-Choose [20, 21] and Commit-and-Prove [22, 23] techniques, which incur significant communication overheads and thus are not suitable for the FL setting. Secret sharing-based solutions as in [24] require secret sharing the inputs, which again induces a huge communication complexity that depends linearly on the model size.

In the privacy-preserving machine learning literature the main focus is on the input privacy of the users. Aside from the secure aggregation techniques [6, 7, 8, 9, 10, 25], another privacy-preserving approach is differential privacy [26]. In FL, employing DP usually entails adding artificial noises to the local models of the users before sending them out for aggregation [27, 28, 29, 30]. None of these works in the DP literature enable computation verification at the users beyond achieving input privacy. We note that clients can still implement local DP in the proposed `LightVeriFL` scheme. In `LightVeriFL` the focus is on the verifiability of the server computations. Even though `LightVeriFL` provides secure verifiable aggregation, it does not verify the integrity of the user inputs. That is, `LightVeriFL` cannot detect the malicious inputs of Byzantine users. In this sense, Byzantine-robust aggregation literature [31, 32, 33, 34] complements the proposed secure verifiable aggregation scheme. It is a great open problem to design a verifiable Byzantine-robust aggregation mechanism, considering various different aggregation rules other than federated averaging.

## B  Overview of Secure Aggregation

In the original FL framework described in Section 3.1, local models of the participating users are received in plain by the server at each iteration. However, these local models carry significant information about the respective users' datasets and using certain attacks, e.g., model inversion, private local data-points of the users can be recovered from their models [2, 3, 4, 5]. To remedy this, authors in [6] propose a secure aggregation scheme, named `SecAgg`, in which the server recovers the aggregate model $\mathbf{y}(t) = \sum_{i \in \mathcal{U}_a(t)} \mathbf{x}_i(t)$ at each iteration $t$ without obtaining any information about the individual local models $\mathbf{x}_i(t) \in \mathcal{U}_a(t)$. In `SecAgg`, the users protect their local models with two separate masks. The first mask is a pairwise mask that is agreed upon between each pair $(i, j)$ of users, $i, j \in [N]$. For this, before the training starts user pair $(i, j)$ agrees on a random seed $a_{i,j}$, where $a_{i,j} = \text{Key.Agree}(sk_i, pk_j) = \text{Key.Agree}(sk_j, pk_i)$. Here, $sk_i$ and $pk_i$ denote the private and public keys of user $i$, respectively. Each user $i$ generates another mask based on a private random seed $b_i$. With these, each user $i$ masks its local model $\mathbf{x}_i$ and sends the masked version $\tilde{\mathbf{x}}_i$ to the server, where[2]

$$\tilde{\mathbf{x}}_\mathbf{i} = \mathbf{x}_i + \text{PRG}(b_i) + \sum_{j:i<j} \text{PRG}(a_{i,j}) - \sum_{j:i>j} \text{PRG}(a_{j,i}). \tag{2}$$

where, PRG stands for a pseudo-random generator. We note that, in (2), the pairwise random masks protect the confidentiality of each user's local model and cancel out at the server upon aggregation. When a user $i$'s local model is only delayed, but not dropped, in order to prevent privacy breaches during this cancellation, the additional mask based on the private seed $b_i$ is used. Each user $i$ secret shares its private seed $b_i$ and private key $sk_i$ with the other users via Shamir's secret sharing [35]. In the aggregation step, the server collects the shares of the dropped users' private keys as well as the shares of the private seeds of the surviving users to reconstruct the pairwise seeds of each dropped

---

[2]Here, we omit the iteration index $t$ for ease of exposition.

user and the private seed of each surviving user, respectively. The server then performs the following to obtain the aggregate model $\mathbf{y}$, which is equal to

$$\sum_{i \in \mathcal{U}_a} \mathbf{x}_i = \sum_{i \in \mathcal{U}_a} (\tilde{\mathbf{x}}_\mathbf{i} - \mathrm{PRG}(b_i)) + \sum_{i \in \mathcal{D}_a} \left( \sum_{j:i<j} \mathrm{PRG}(a_{i,j}) - \sum_{j:i>j} \mathrm{PRG}(a_{j,i}) \right), \qquad (3)$$

where $\mathcal{U}_a$ and $\mathcal{D}_a$ represent the set of surviving and dropped users at the aggregation stage, respectively. The `SecAgg` scheme guarantees local model privacy as long as the number of dropped users at aggregation $D_a$ and the number of colluding users $T$ satisfy $N - D_a > T$.

A major performance bottleneck of the `SecAgg` scheme is the fact that it requires the server to reconstruct the seeds one by one (the private seeds of the surviving users as well as the pairwise seeds of the dropped users), which incurs a computation overhead of $O(N^2)$. To remedy this, more efficient secure aggregation schemes have been proposed [7, 8]. In [8], authors propose `LightSecAgg`, in which the server recovers the aggregate mask of the surviving users in one-shot, thus overcoming the aforementioned $O(N^2)$ bottleneck. In `LightSecAgg`, users still protect their models with local masks but these masks are encoded and shared with the other users such that once the server receives sufficiently many responses from the users, it can reconstruct the aggregate mask of these surviving users and hence recover the aggregate model. The approach in `LightSecAgg` constitutes the basis of our approach in designing a lightweight and verifiable aggregation protocol.

**Limitations of the secure aggregation schemes.** With secure aggregation alone, users cannot verify whether i) the aggregation result provided by the server is correct and ii) their individual models are accounted for in the aggregation. Thus, without a verifiable aggregation scheme, users are vulnerable to a malicious/lazy server forging the aggregation result and/or not incorporating local models of all users.

## C   Overview of the Baseline Protocol: `VeriFL`

In this section, we give an overview of the existing verifiable aggregation protocols for FL. The first work on verifiable aggregation is [11]. In that work, authors propose `VerifyNet` that utilizes a zero knowledge proof technique such that upon aggregation, the server sends a proof to the users indicating the correctness of the aggregation. Users may accept or reject this proof. The major bottleneck in this system is the fact that the size of this proof increases linearly with $d$, the dimension of the model. Thus, the time required for verification using the zero knowledge proof increases linearly with $d$, making the `VerifyNet` scheme impractical for real-life learning models with large number of parameters.

Reference, [12] proposes a verifiable aggregation scheme named `VeriFL`, which is the baseline scheme we consider in this work. The `VeriFL` scheme utilizes homomorphic hash functions of the local models as well as commitments to design a communication-efficient verification scheme. In `VeriFL`, the communication overhead is independent of $d$, thus making the `VeriFL` scheme more efficient than `VerifyNet`. Below we give a brief overview of the `VeriFL` scheme.

In `VeriFL`, training iterations are performed as described in Section 3.1. The `VeriFL` scheme starts with a preparation stage. Before sending out its updated local model, each participating user $i$ generates a linearly homomorphic hash $h_i$ of its local model $\mathbf{x}_i$. Based on this hash $h_i$, user $i$ then generates a commitment $c_i$ using a commitment scheme COM such that $c_i = \mathrm{COM.Commit}(h_i, r_i)$, where $r_i$ is a uniformly random string privately sampled by user $i$. Next, each user $i$ forwards its commitment string $c_i$ to all the other participating users. Once this step is completed for all user pairs, each user sends its local model to the server. In `VeriFL`, in sending the local models to the server for aggregation, users follow the `SecAgg` protocol described in Appendix B. Once the server recovers the aggregate model $\mathbf{y}$ and pushes it to the users, the verification stage of the `VeriFL` scheme commences with the decommitment step, in which each user $i$ receives the decommitment strings $(h_j, r_j)$ from all the other users $j \in [N]$ to check if $c_j$ received in the preparation step satisfies

$$c_j = \mathrm{COM.Commit}(h_j, r_j). \qquad (4)$$

If (4) is not satisfied for at least one other user $j$, user $i$ terminates the process and regards the aggregation $\mathbf{y}$ as forged. If no issue is detected at this step for any of the users, user $i$ proceeds and

checks the equality of the hashes. For this, user $i$ first computes the hash of the aggregate model, $h_{\mathrm{agg}}$ and then check if the following holds

$$h_{\mathrm{agg}} = \prod_{i=1}^{N} h_i. \tag{5}$$

If (5) does not hold, user $i$ raises a flag and regards the aggregate result as forged, otherwise user $i$ accepts the result and starts generating its updated local model for the next iteration.

We note that thanks to `SecAgg`, the above `VeriFL` scheme can tolerate dropouts during aggregation. However, dropouts occurring during the verification step need to be handled to successfully verify the integrity of the aggregation. For this, each user $i$ secret shares its decommitment string $(h_i, r_i)$ with the other users in the preparation step. In the verification step, after decommitting, surviving users send the shares they have received from the dropped users to the server, which recovers the decommitment strings $(h_j, r_j)$ of the dropped users and sends them back to the surviving users. Equipped with these, along with the decommitment strings of the other surviving users, each surviving user $i$ performs (4) and (5).

**Limitations of the `VeriFL` scheme in [12].** i) As in the original `SecAgg` scheme, the `VeriFL` scheme suffers from the one-by-one reconstruction of the dropped users' decommitment strings $(h_j, r_j)$, which are then used by the surviving users in performing the verification through (4) and (5). This one-by-one reconstruction at the server incurs an $O(N^2)$ computation bottleneck and significantly slows down the system for large $N$. For example, when $N = 500$ and $d = 100K$, when the dropout rate is %30, while the actual verification steps in (4) and (5) take around 5 seconds (majority of which is the hash computation of the aggregate model), the reconstruction of the dropped decommitment strings takes approximately 150 seconds, incurring a significant burden on the verification procedure. ii) In `VeriFL`, the homomorphic hashes of the honest users are revealed in the verification phase (in decommitment for surviving users and during the reconstruction at the server for the dropped users). Since the homomorphic hash of a model vector is a deterministic function of the inputs, an adversary may use the revealed hash result to guess the local model of an honest user. Thus, in `VeriFL`, the local model confidentiality of the honest users may be broken, particularly when the gradient vector has only a few non zero entries [36].

Motivated by these limitations, in this work, we propose `LightVeriFL`, which forgoes one-by-one reconstruction of the dropped users' decommitment strings $(h_j, r_j)$. Instead, in `LightVeriFL`, the server performs a one-shot reconstruction of the aggregate decommitment strings of the dropped and surviving users. By this way, not only we avoid the major $O(N^2)$ computation bottleneck in reconstruction, thus making the verification significantly faster but also avoid revealing the individual hashes of any of the users (dropped or surviving) to any of the parties (users and the server), circumventing the aforementioned privacy breach.

## D   Detailed Description of `LightVeriFL`

In this section, we describe the `LightVeriFL` scheme in detail. We require the elements of the local models of the users and the aggregate model to lie in $\mathbb{F}_q^d$, $q$ is a prime number.[3][4] The key feature of the `LightVeriFL` scheme is that the server is able to reconstruct the aggregate hashes of the dropped users in one-shot as opposed to `VeriFL` [12], which has the server reconstructing the dropped hashes one-by-one. In `LightVeriFL`, the server additionally recovers the aggregate hash of the surviving users so that these surviving users never reveal their hashes in plain to other users.

The proposed `LightVeriFL` scheme has two phases: aggregation phase and verification phase. In the aggregation phase, additional operations are performed on top of `SecAgg` to enable verification later on. The verification phase happens after the users receive the aggregate model from the server. In order to protect the privacy of the local models, in `LightVeriFL`, users mask their hashes before sending them to the server. Then, in the verification phase, they make use of the additive homomorphism of the constructed hashes described in Section 2 to verify the integrity of the aggregation. That is, after

---

[3]We select a large $q$ such that the field $\mathbb{F}_q^d$ is large enough to avoid any wrap-around during aggregation.

[4]We assume that each user $i$ converts its model $\mathbf{x}_i$ from real domain to finite field $\mathbb{F}_q$ through quantization at each iteration before invoking `LightVeriFL`. There exists quantization schemes in the literature that ensure convergence of the global model [8].

receiving the aggregation, each user needs to be able to check whether the following holds:

$$h(\mathbf{y}) = \prod_{i \in \mathcal{U}} h_i(\mathbf{x}_i) \prod_{j \in \mathcal{D}} h_j(\mathbf{x}_j), \tag{6}$$

where $\mathbf{y}$ is the aggregate model obtained from the server and $\mathcal{U}$ and $\mathcal{D}$ show the set of surviving and dropped users during verification, respectively. As mentioned earlier, in `LightVeriFL`, unlike `VeriFL` [12], we do not want the server to reconstruct the hashes one-by-one. Then, in order for users to perform (6), the server needs to reconstruct the product of the hashes of the surviving and dropped users as in `LightVeriFL` no user sees another user's hash in plain. This is when elliptic curves comes into play, which have the following property.

**Property 1.** The order of a point $P$ on the EC is the smallest integer $n$ such that $nP = \mathcal{O}$. Given $E(\mathbb{F}_p)$, one can find a base point $P$ that generates a cyclic subgroup of order $n$, where $n < p$ is a prime number. That is, if $k$ and $\ell$ are integers $kP = \ell P$ if and only if $k \equiv l$ in modulo $n$ [15].

By utilizing this cyclic subgroup property, we perform all encoding and decoding operations in `LightVeriFL` on ECs with modulus $n$. When the linearly homomorphic hash described in (1) is implemented using an EC[5], the condition in (6) becomes

$$h(\mathbf{y}) = \sum_{i \in \mathcal{U}} h_i(\mathbf{x}_i) + \sum_{j \in \mathcal{D}} h_j(\mathbf{x}_j), \tag{7}$$

using the EC analogues of the operations in multiplicative groups. That is, in this case the server needs to recover the aggregate hashes of the users analogous to the secure aggregation problem [6, 8]. Thus, when the hashes are constructed over the EC, surviving users need the aggregation of the hashes of all users to perform the verification step in (7). In this case, existing secure aggregation tools become applicable to the problem at hand. We note that we implement a variant of the Pedersen commitment scheme on the EC such that $c = \alpha h + \beta r$, where $h$ is hash of a user, denoted by a point on the EC, whereas $r$ is a randomly chosen point on the curve. Here, $\alpha$ and $\beta$ are integer coefficients agreed upon by the users before the start of the protocol. Once the commitments are implemented over the EC, the additive homomorphic property described in Section 2 becomes $c(h_1 + h_2, r_1 + r_2) = c(h_1, r_1) + c(h_2, r_2)$.

Formally, the `LightVeriFL` scheme assumes that out of the $N$ users, at most $T$ of them colludes with each other while $D$ users drops during the verification phase of the `LightVeriFL` scheme.[6] Here, we have $0 \leq T \leq N - 1$ and $0 \leq D \leq N - 1$. We let $U$ denote the targeted number of surviving users during the verification step where $N - D \geq U = T + 1$. We detail the two phases of the `LightVeriFL` scheme next. The pseudo code of `LightVeriFL` is given in Algorithm 1 in Appendix E.

**1. The Aggregation Phase:** In order to verify the aggregation computed by the server, users perform the following additional operations during the aggregation phase in addition to the implemented secure aggregation protocol, i.e., `SecAgg`.

**Round A.0: Advertising Keys.** `LightVeriFL` starts with the setup operations of the linearly homomorphic hash and commitment schemes, e.g., fixing an EC with generator $g$ and subgroup order $n$, setting up $d$ distinct points on the EC for the hash computation in (1), and so on. The operations in this round do not depend on the local models of the users. For the `SecAgg` protocol, during this phase users agree on pairwise masks as well as described in Appendix B.

**Round A.1: Offline Encoding and Sharing Local Masks.** This step is inspired by the `LightSecAgg` scheme in [8] except that all the masks are sampled from an EC, i.e., masks are points on the curve. First each user $i \in [N]$, generates a mask $z_i$, which is a point on the EC. Next, user $i$ randomly samples jointly uniform points $[n_i]_k$ from the EC for $k \in \{2, \ldots U\}$. Using these randomly picked $[n_i]_k$, user $i$ encodes the mask $z_i$ as follows

$$[\tilde{z}_i]_j = (z_i, [n_i]_2, \ldots, [n_i]_U) \cdot W_j. \tag{8}$$

---

[5]When the hash is implemented over an EC, hash of each user becomes a point on that EC.

[6]We assume that there is no dropout in the aggregation stage as in the verification problem the critical dropouts happen when users whose results were aggregated in the model update drop at the time of the verification. We note that existing secure aggregation schemes take care of the dropouts occurring during the aggregation phase and the proposed `LightVeriFL` scheme can be implemented on top of these secure aggregation schemes to tolerate dropouts both in aggregation and verification phases.

Here, $W_j$ is the $j$th column of the $T$-private MDS matrix $W \in \mathbb{F}_n^{U \times N}$, where $n$ is the order of the subgroup on EC. The use of matrix $W$ in encoding protects the generated masks from any subset of colluding $T$ users. We can always generate such $T$-private MDS matrix for a given $U$, $N$, and $T$ [35, 37, 8]. After the encoding, each user $i \in [N]$, sends its encoded mask $[\tilde{z}_i]_j$ to user $j \in [N] \backslash \{i\}$ so that in the end of this step, each user $i \in [N]$ has $[\tilde{z}_j]_i$ from all users $j \in [N]$.

**Round A.2: Hash Generation and Uploading Masked Hashes.** In this step, each user $i \in [N]$ updates its local model $\mathbf{x}_i$ and then based on this model computes its hash $h_i$[7] according to (1). Next, each user computes its commitment $c_i$ based on its hash according to the aforementioned variant of the Pedersen commitment. In generation of the commitment, each user uniformly samples a point $r_i$ on the EC to be used in decommitting. Each user $i \in [N]$ sends its commitment $c_i$ to users $j \in [N] \backslash \{i\}$ such that at the end of this step, each user has commitments of every other user. Once the commitment exchange is completed, each user $i \in [N]$ uploads its masked hash $\tilde{h}_i = h_i + z_i$ and masked randomness $\tilde{r}_i = r_i + z_i$ to the server along with its masked local model that is generated according to the employed secure aggregation scheme.

**Round A.3: Aggregate Model Recovery.** The server recovers the aggregate model $\mathbf{y}$ and sends it back to the users.

This concludes the `LightVeriFL` operations during the aggregation phase.

**2. Verification Phase:** Having received the aggregate model $\mathbf{y}$ at the end of the aggregation phase, users perform the verification phase to check the integrity of the aggregation.

**Round V.0: Aggregate Decommitting.** In this step, the server performs one-shot recovery of the aggregate decommitment strings $(h_i, r_i)$ of the dropped and surviving users. As mentioned earlier, we let $\mathcal{U}$ and $\mathcal{D}$ denote the set of surviving and dropped users in the verification stage, respectively. In order to recover the aggregate mask of the surviving users, $\sum_{i \in \mathcal{U}} z_i$, each surviving user $i$ is notified to send the aggregate encoded mask it has received from other surviving users, $\sum_{j \in \mathcal{U}} [\tilde{z}_j]_i$. Upon receiving $U$ such messages, the server is able to decode the aggregate mask of the surviving users $\sum_{i \in \mathcal{U}} z_i$ due to the MDS property of the encoding in (8). Next, a similar one-shot decoding is repeated for the aggregate mask of the dropped users and the server reconstructs $\sum_{k \in \mathcal{D}} z_k$. Finally, the server is able to recover the aggregate decommitment strings of the surviving and users with

$$\sum_{i \in \mathcal{U}} h_i = \sum_{i \in \mathcal{U}} \tilde{h}_i - \sum_{i \in \mathcal{U}} z_i \tag{9}$$

$$\sum_{i \in \mathcal{U}} r_i = \sum_{i \in \mathcal{U}} \tilde{r}_i - \sum_{i \in \mathcal{U}} z_i. \tag{10}$$

The same one-shot recovery steps in (9)-(10) are performed for the decommitment strings of the dropped users in $\mathcal{D}$ as well.[8] The server sends these decommitment strings back to the surviving users. The next step of the users is to verify the integrity of the reconstructions performed by the server. For this, we utilize the homomorphic additive property of the described commitment scheme such that each user checks

$$c \left( \sum_{i=1}^N h_i, \sum_{i=1}^N r_i \right) = \sum_{i=1}^N c_i, \tag{11}$$

using the commitments received from the users in Round A.1 during the aggregation phase. In (11) we have $\sum_{i=1}^N h_i = \sum_{i \in \mathcal{U}} h_i + \sum_{k \in \mathcal{D}} h_k$. If (11) does not hold for a user, then that user raises a flag and rejects the aggregate hash recovery performed by the server. Otherwise, users proceed with the next round.

**Round V.1: Batch Checking.** Having accepted the reconstructed hash results received from the server, in this round users verify the correctness of the aggregation result $\mathbf{y}$. First, each user computes

---

[7]Here, we denote user $i$'s hash simply with $h_i$ instead of $h_i(\mathbf{x}_i)$ and leave the dependence on $\mathbf{x}_i$ implicit.

[8]In secure aggregation schemes, the server performs key/mask reconstructions either for the dropped users [6] or for the surviving users [8]. In our verification problem, the server reconstructs the aggregate masks of all users (surviving and dropped) in order to avoid decommitment string exchange among the surviving users in the verification stage so that no individual user observes the decommitment string of another user to protect the input privacy of the users.

the hash of the aggregate model $\mathbf{y}$, denoted by $h_{agg}$, using the construction in (1) over the EC. Next, each user checks if the following condition is satisfied:

$$h_{agg} = \sum_{i=1}^{N} h_i. \tag{12}$$

If the condition in (12) is satisfied, then users accept the aggregated model and proceed with the next iteration of training. Otherwise, they regard the aggregate model computed by the server as forged and reject the result.

**Improved `LightVeriFL` with Amortized Verification:** In the amortized verification, Round V.0 of the verification phase stays the same. On the other hand, Round V.1 of the verification phase is only performed at every $L$ iterations, where $L$ is the predetermined batch size of the verification protocol. That is, during iterations $\ell \in [L]$, each user samples a random coefficient $\alpha_\ell$ and stores the sum of the hashes of the users reconstructed by the server as $h^\ell = \sum_{i=1}^{N} h_i^\ell$ as well as the aggregate model in the $\ell$th iteration $\mathbf{y}(\ell)$. Then, in the $L$th iteration, it checks if the following relationship holds:

$$h\left(\sum_{\ell \in [L]} \alpha_\ell \mathbf{y}(\ell)\right) = \sum_{\ell \in [L]} \alpha_\ell h^\ell. \tag{13}$$

If (13) holds for a user, then that user verifies the aggregations of the past $L$ iterations all at once by performing only one hash computation in verification as opposed to performing $L$ expensive hash computations during verification, one for each of the $L$ iterations.

## E  Pseudo Code of `LightVeriFL`

The pseudo code of `LightVeriFL` is given in Algorithm 1.

## F  Proofs of Theoretical Guarantees

**Theorem 1.** *[Correctness of Verification] Under the `LightVeriFL` scheme, users accept the aggregation results $\mathbf{y}(\ell)$, $\ell \in [L]$ if and only if these results are correctly aggregated by the server with probability (almost) 1.*

*Proof.* In the final amortized verification step, as described earlier, each user finds the hash of the weighted sum of the aggregate models from last $L$ iterations, i.e., $\sum_{\ell \in [L]} \alpha_\ell \mathbf{y}(\ell)$. During the verification, the server acts alone without colluding with the users. During the entire execution of `LightVeriFL`, users receive two computation results from the server. The first one is the aggregate hash reconstruction of the surviving and dropped users. If the server sends incorrectly reconstructed aggregate hashes in this step, during the aggregate commitment check step in (11) users detect the error in the aggregate hash reconstruction and regard the result as forged.

Assuming that the reconstructed hashes are accepted by the users, the second computation to be verified by the users is the aggregate models $\mathbf{y}(\ell)$, $\ell \in [L]$. Let us assume, as in [12], that the server successfully spoofs the aggregation result in $\mathcal{K}$ iterations, $\mathcal{K} \subseteq [L]$ and sends $\bar{\mathbf{y}}(k)$ instead of $\mathbf{y}(k)$, with $\mathbf{y}(k) \neq \bar{\mathbf{y}}(k)$ for $k \in \mathcal{K}$. Then, for the users to accept the incorrectly aggregated results, for the following needs to be satisfied

$$\sum_{i=1}^{d} \sum_{k \in \mathcal{K}} \alpha_k \mathbf{y}(k)[i] g_i = \sum_{i=1}^{d} \sum_{k \in \mathcal{K}} \alpha_k \bar{\mathbf{y}}(k)[i] g_i, \tag{14}$$

where $\mathbf{y}(k)[i]$ denotes the $i$the element of the aggregated model, $i \in [d]$. We note that $\alpha_k$s are uniformly selected numbers by each user from elliptic curve subgroup of order $n$ such that the following condition holds

$$\sum_{k \in \mathcal{K}} \alpha_k \mathbf{y}(k)[i] = \sum_{k \in \mathcal{K}} \alpha_k \bar{\mathbf{y}}(k)[i], \tag{15}$$

with probability $\frac{1}{n}$, even when $\mathbf{y}(k) \neq \bar{\mathbf{y}}(k)$ for $k \in \mathcal{K}$, $i \in [d]$. Here, since the elliptic curves are designed to have a large subgroup order $n$, this event is negligible.[9]  With that in mind, the

---

[9]In the NIST P-256 curve we use in our implementation, the subgroup order $n$ is a 256-bit number [16].

**Algorithm 1** The `LightVeriFL` protocol

**Input:** $T$ (privacy guarantee), $D$ (dropout-resiliency guarantee), $U = T + 1$ (target number of surviving users during the verification), Round A.0: Advertising Keys takes place before the protocol starts.

1: **Server Executes:**
2: *// Aggregation Phase*
3: *// Round A.1: Offline Encoding and Sharing Local Masks*
4: **for** each user $i = 1, 2, \ldots, N$ **in parallel do**
5:     $z_i \leftarrow$ randomly selected point from an elliptic curve (EC) of subgroup order $n$
6:     $[n_i]_2, \ldots, [n_i]_U \leftarrow$ randomly selected points from the same EC of subgroup order $n$
7:     $\{[\tilde{z}_i]_j\}_{j \in [N]} \leftarrow$ obtained by encoding $z_i$ and $[n_i]_k$'s as in (8)
8:     sends encoded mask $[\tilde{z}_i]_j$ to user $j \in [N] \backslash \{i\}$
9:     receives encoded mask $[\tilde{z}_j]_i$ from user $j \in [N] \backslash \{i\}$
10: **end for**
11: *// Round A.2: Hash Generation and Uploading Masked Hashes*
12: **for** each user $i = 1, 2, \ldots, N$ **in parallel do**
13:     *// user $i$ obtains $\mathbf{x}_i$ after the local training*
14:     generates hash $h_i$ based on $\mathbf{x}_i$ according to (1) on the same EC
15:     $r_i \leftarrow$ randomly selected point from the same EC of subgroup order $n$
16:     generates commitment $c_i = \text{COM.Commit}(h_i, r_i)$, the COM scheme is described in Appendix D
17:     sends commitment $c_i$ to user $j \in [N] \backslash \{i\}$
18:     receives commitment $c_j$ from user $j \in [N] \backslash \{i\}$
19:     $\tilde{h}_i \leftarrow h_i + z_i$ and $\tilde{r}_i \leftarrow r_i + z_i$ *// masks its hash and randomness*
20:     uploads masked hash and randomness $\tilde{h}_i, \tilde{r}_i$ to the server
21: **end for**
22: *// Round A.3: Aggregate Model Recovery*
23: recovers the aggregate model $\mathbf{y}$ and sends it to the users
24: **for** each user $i = 1, 2, \ldots, N$ **in parallel do**
25:     receives the aggregate model $\mathbf{y}$ from the server
26: **end for**
27: *// Verification Phase*
28: identifies set of surviving users $\mathcal{U} \subseteq [N]$ in the verification phase
29: *// Round V.0: Aggregate Decommitting*
30: **for** each user $i \in \mathcal{U}$ **in parallel do**
31:     computes aggregated encoded masks $\sum_{j \in [N]} [\tilde{z}_j]_i$
32:     uploads aggregated encoded masks $\sum_{j \in [N]} [\tilde{z}_j]_i$ to the server
33: **end for**
34: collects $U$ messages of aggregated encoded masks $\sum_{j \in [N]} [\tilde{z}_j]_i$ from user $i \in \mathcal{U}$
35: *// recovers the aggregate mask*
36: $\sum_{i \in [N]} z_i \leftarrow$ obtained by decoding the received $U$ messages
37: *// recovers the aggregate hash and randomness of the users*
38: $\sum_{i \in [N]} h_i \leftarrow \sum_{i \in [N]} \tilde{h}_i - \sum_{i \in [N]} z_i$ and $\sum_{i \in [N]} r_i \leftarrow \sum_{i \in [N]} \tilde{r}_i - \sum_{i \in [N]} z_i$
39: sends $\sum_{i \in [N]} h_i$ and $\sum_{i \in [N]} r_i$ to $\mathcal{U}$
40: **for** each user $i \in \mathcal{U}$ **in parallel do**
41:     receives $\sum_{i \in [N]} h_i$ and $\sum_{i \in [N]} r_i$ from the server
42:     *// verifies the integrity of $\sum_{i \in [N]} h_i$ and $\sum_{i \in [N]} r_i$*
43:     checks $\text{COM.Commit}\left(\sum_{i \in [N]} h_i, \sum_{i \in [N]} r_i\right) \overset{?}{=} \sum_{i \in [N]} c_i$
44:     if above is true, accepts the aggregate hash and randomness recovered by the server and moves to the next round
45:     otherwise, regards the result as forged
46: **end for**
47: *// Round V.1: Batch Checking*
48: **for** each user $i \in \mathcal{U}$ **in parallel do**
49:     computes the hash of the aggregate model $\mathbf{y}$ denoted by $h_{agg}$ using (1) on the same EC
50:     *// verifies the integrity of the aggregation*
51:     checks $h_{agg} \overset{?}{=} \sum_{i \in [N]} h_i$
52:     if above is true, accepts the aggregate model and performs local training for the next training round
53:     otherwise, regards the aggregation result as forged
54: **end for**

condition in (14) does not hold due to the collision resistance property of the linearly homomorphic hashes we use as described in (1). That is, no two different vectors produce the same hash with overwhelming probability such that the aggregate hash verification step in (12) detects any spoofing attempts launched by the server.

Thus, given the server acts independently to forge the aggregation results, the aggregate commitment check in (11) along with the collision property of the hash guarantees verification of the aggregation results provided by the server. □

**Theorem 2.** *[Input Privacy Guarantee] The proposed* `LightVeriFL` *protocol provides input privacy against up to* $T$ *colluding users.*

*Proof.* The use of a secure aggregation scheme such as `SecAgg` [6] or `LightSecAgg` [8] takes care of the input privacy in the aggregation protocol. Additionally, we need show that hashes and commitments used in the `LightVeriFL` protocol do not violate input privacy of the users.

As discussed in [36], if revealed, the linearly homomorphic hashes used in verification schemes can help an adversary guess the input vector of a user, particularly if the input vector has a few non-zero elements. That is, when the hash of a user is revealed, the distinguisher can tell the difference between the simulated case and the actual case. Thus, in our scheme, no party (including the server and up to $T$ colluding users) has access to the hash of an honest user. More formally, the proposed `LightVeriFL` protocol guarantees the following mutual information condition given in (16) for an arbitrary set $\mathcal{T}$

$$I\left(\{h_i\}_{i\in[N]}; \{h_i + z_i\}_{i\in[N]}, \left\{\sum_{j\in[N]}[\tilde{z}_j]_i\right\}_{i\in\mathcal{U}} \middle| \sum_{i\in[N]} h_i, \{h_i\}_{i\in\mathcal{T}}, \{z_i\}_{i\in\mathcal{T}}, \{[\tilde{z}_j]_i\}_{j\in[N],i\in\mathcal{T}}\right) = 0,$$

(16)

of $T$ colluding users and a surviving user set $\mathcal{U}$ such that $\mathcal{U} \subseteq [N], |\mathcal{U}| \geq U, U = T + 1$. In order to show (16) we use similar steps as in [8, Proof of Theorem 1].

In addition to the hashes, we need to consider the commitments exchanged during the execution of `LightVeriFL` in the Round A.2 of the aggregation phase as described in Appendix D. During this round, each user receives the commitments of all the other users. The commitment scheme we use is perfectly hiding. That is even though an adversarial user, given that it has enough compute resources, can find multiple decommitment pairs leading to the same commitment, that adversarial user has no way of determining the actual hash of another user from the received commitment, beyond random guessing. Thus, exchanging commitments does not violate the input privacy of users. □

**Theorem 3.** *[Dropout Resilience in Verification] The proposed* `LightVeriFL` *scheme guarantees dropout resilience up to any* $D$ *dropped users during the verification phase such that* $N \geq T + D + 1$.

*Proof.* In `LightVeriFL`, each user $i$ encodes its mask $z_i$ using the same $T-$private MDS matrix such that the aggregate encoded mask that a surviving user $j$ sends to the server in aggregate decommitting satisfies

$$\sum_{i\in\mathcal{D}}[\tilde{z}_i]_j = \left(\sum_{i\in\mathcal{D}} z_i, \sum_{i\in\mathcal{D}}[n_i]_2, \ldots, \sum_{i\in\mathcal{D}}[n_i]_U\right) \cdot W_j,$$

(17)

where $W_j$ is the $j$th column of the MDS matrix $W$. From (17), we see that $\sum_{i\in\mathcal{D}}[\tilde{z}_i]_j$ is the encoded version of the desired aggregate mask (of the dropped users) $\sum_{i\in\mathcal{D}} z_i$. By construction, we have $N - D \geq U$ so that the server receives $\sum_{i\in\mathcal{D}}[\tilde{z}_i]_j$ from at least $U$ users. Thus, through MDS decoding, the server successfully recovers the aggregate mask of the dropped users $\sum_{i\in\mathcal{D}} z_i$ and performs $\sum_{i\in\mathcal{D}} h_i = \sum_{i\in\mathcal{D}}(h_i + z_i) - \sum_{i\in\mathcal{D}} z_i$. A similar encoding/decoding strategy holds for the aggregate mask of the surviving users $\sum_{i\in\mathcal{U}}[\tilde{z}_i]_j$. Thus, the server is able to reconstruct the aggregate hashes of the users when up to $D$ users drop where $N - D \geq U = T + 1$. Using the aggregate hash of the users, the surviving users perform the verification operation even in the presence of dropouts. □

**Theorem 4.** *The proposed `LightVeriFL` scheme guarantees successful aggregation integrity verification in the presence of any $D$ user dropouts during the verification phase without sacrificing input privacy against up to any $T$ colluding users for $T + D < N$. When `LightVeriFL` is implemented together with a secure aggregation scheme, such as `SecAgg` [6] and `LightSecAgg` [8], a secure verifiable aggregation scheme is obtained.*

Theorem 4 simply follows from Theorems 1-3.We note that in `LightVeriFL`, as in the secure aggregation schemes, the guarantee in Theorem 4 is for a single FL iteration only. Since all the randomness in `LightVeriFL` is generated independently across all iterations, this guarantee can be extended to the entire FL protocol by invoking the proposed `LightVeriFL` scheme at each iteration.

## G    Complexity Analysis

In this section, we provide the complexity analysis of the `LightVeriFL` scheme for $N - D \geq U = T + 1$ as in [8]. We do not include the complexity of the `SecAgg` scheme as our `LightVeriFL` can be implemented in a standalone manner without any secure aggregation protocol. We note that the hashes and commitments have constant lengths that are independent of the model size $d$ and $N$.

**Offline storage cost.** In `LightVeriFL`, each user generates a random mask $z_i$ (a point on the EC) of constant length and stores coded masks of all the other users as well as their constant-length commitments. In addition, in the amortized verification, each user stores the past $L$ aggregate models. Thus, the total storage cost of `LightVeriFL` at each user is $O(N + Ld)$.

**Offline communication and computation loads.** In `LightVeriFL`, each user computes its coded masks in offline manner before the local model is computed. To compute the coded mask, each user performs an $(N, U)$ MDS coding, where the size of each data block is constant (since $z_i$ and $[n_i]_k$, $k = \{2, \ldots, U\}$ are all points on the EC). Thus, the offline computation load at each user at each iteration is $O(N \log N)$. Then, still in the offline mode, each user shares each of the $N$ coded segments with the other users, which induces a communication load of $O(N)$.

**Online communication load.** Each user sends its masked hash to the server as well as the corresponding commitment to the other users during the aggregation phase. Since either hash and commitment has a constant length, the total communication load for the users during the aggregation phase is $O(N)$. In addition, in the verification phase, each surviving user sends the aggregate coded masks it has received from the other users (which has a constant length) to the server. Thus, the total online communication load for a user is $O(N)$. Correspondingly, the online communication load at the server is $O(N + U)$ since it receives the masked hashes from $N$ users in the aggregation phase as well as the coded masks from the surviving $U$ users in the verification phase.

**Online computation load.** After computing its local model, each user computes its hash, which is the most expensive operation in `LightVeriFL` at the users and introduces an $O(d)$ computation load, $d$ is the model size. Then, in the amortized verification each surviving user computes the hash of the aggregate model at every $L$ iterations, where $L$ is the batch size, inducing another $O(\frac{d}{L})$ computation load, making the total online computation load $O(\frac{L+1}{L}d)$ at each user The server, on the other hand, performs the reconstruction from the coded masks it has received from $U$ surviving users to recover $\sum_{i \in \mathcal{U}} z_i$ and $\sum_{i \in \mathcal{D}} z_i$, which is the most time-consuming operation performed at the server in `LightVeriFL`. For this reconstruction, the server decodes a $U$ dimensional MDS code using $U$ coded messages it has received from the surviving users. Since each message is of a constant length, the total computation complexity here is $O(U \log U)$ operations in $\mathbb{F}_n$, where $n$ is the subgroup order of the EC.

Next, we compare the complexity of the proposed `LightVeriFL` with the baseline `VeriFL` [12] in Table 2. We consider a scenario, where $T = \frac{N}{2}$, $U = T + 1$, and $D = pN$, where $0 \leq p < \frac{1}{2}$. From Table 2, we see that in the proposed `LightVeriFL` scheme, the recovery complexity at the server is almost linear in the number of users $N$, whereas in the `VeriFL` scheme reconstruction has quadratic complexity in $N$, as in `VeriFL` the server reconstructs each of the dropped users' hashes one-by-one. Thus, `LightVeriFL` significantly improves the computation time at the server, thereby speeding up the entire verification process especially when the number of users $N$ grows. Further, in `LightVeriFL`, the masks that are used to hide the hashes are prepared in advance in an offline manner as these masks are independent of the local models as well as the hashes. In `VeriFL`, however, to tolerate dropouts, generated hashes are secret shared among the users, which can be performed in

Table 2: Per iteration complexity comparison of the standalone implementations of `VeriFL` [12] and the proposed `LightVeriFL`. Here, $N$ is the number of users, $d$ is the model size. In this table, we use S to denote the server and U to denote a user.

|  | VeriFL | LightVeriFL |
|---|---|---|
| offline comm. (U) | $-$ | $O(N)$ |
| offline comp. (U) | $-$ | $O(N \log N)$ |
| online comm. (U) | $O(N)$ | $O(N)$ |
| online comm. (S) | $O(N^2)$ | $O(N)$ |
| online comp. (U) | $O(N^2 + \frac{L+1}{L}d)$ | $O(\frac{L+1}{L}d)$ |
| reconstruction (S) | $O(N^2)$ | $O(N \log N)$ |

Table 3: Per iteration complexity comparison of `SecAgg` [6] and the proposed `LightVeriFL`. Here, we do not include a secure aggregation scheme in `LightVeriFL` (only verification-related operations are considered) and $N$ is the number of users, $d$ is the model size, $s$ is the length of the secret keys in `SecAgg`, $s << d$. In this table, we use S to denote the server and U to denote a user.

|  | SecAgg | LightVeriFL |
|---|---|---|
| offline comm. (U) | $O(sN)$ | $O(N)$ |
| offline comp. (U) | $O(dN + sN^2)$ | $O(N \log N)$ |
| online comm. (U) | $O(d + sN)$ | $O(N)$ |
| online comm. (S) | $O(dN + sN^2)$ | $O(N)$ |
| online comp. (U) | $O(d)$ | $O(\frac{L+1}{L}d)$ |
| reconstruction (S) | $O(dN^2)$ | $O(N \log N)$ |

an online manner after the local training is completed. Thus, the online computation complexity at the users and the online communication complexity at the server is higher than those of `LightVeriFL`, which does not require secret sharing among the users and one-by-one reconstruction at the server.

We compare the complexities of standalone implementations of `SecAgg` [6] and our proposed `LightVeriFL` scheme in Table 3. We observe from Table 3 that the complexity of standalone implementation of `LightVeriFL` is not significant compared to the `SecAgg` scheme, as in `LightVeriFL`, hashes and commitments are of constant length independent of $d$. That is, when the `LightVeriFL` protocol is implemented on top of `SecAgg` [6], the complexity of the overall scheme is no more than `SecAgg` alone. Thus, achieving a secure verifiable aggregation scheme through `LightVeriFL` is not infeasible in practical scenarios, where `SecAgg` is implemented. A similar argument applies to more efficient secure aggregation schemes such as `SecAgg+` [7] and `LightSecAgg` [8].

## H   Experimental Details and Additional Experiment Results

In our implementation, we use the open-source `fastecdsa` Python library for fast elliptic curve cryptography [38]. The implementation of the proposed `LightVeriFL` scheme can be found in `https://github.com/bbuyukates/LightVeriFL-fast`.

In Tables 1 and 4, we use $L = 10$ for the batch size of the amortization and report the average of $5$ independent runs along with the standard error. In Table 1, we observe that in `LightVeriFL` decommitting round is not affected in all three dropout scenarios, since the server reconstructs the aggregate hash of all $N = 200$ users in one-shot whereas in the `VeriFL` scheme the time needed for decommitting increases as more and more users drop the protocol. The batch checking round in both of the schemes is not affected by varying dropout rates aside from a minor straggler effect. That is, as the dropout rate decreases, more number of users stay in the system for verification, which in turn increases the completion time of the verification phase due to the straggler effect.

To report the performance results for realistic FL settings we show the performance of `LightVeriFL` for varying $d$ as well. For this, we use $d = 10K$ (similar to a logistic regression model on MNIST [39] which requires $d = 7,850$) and $d = 1M$ (similar to training a CNN [1] on FEMNIST [40], which requires $d = 1,206,590$).

Table 4: Breakdowns of the verification time of `LightVeriFL` and `VeriFL` for model size $d$ with $N = 200$ users, 30% dropout rate, and $L = 10$. All times are in seconds.

|  | Phase | d=10K | d=100K | d=1M |
|---|---|---|---|---|
| | Round V.0 Decommitting | 29.72 | 30.56 | 35.44 |
| VeriFL | Round V.1 Batch Checking | 0.62 | 6.03 | 60.27 |
| | Verification Phase - Total | $30.34 \pm 0.40$ | $36.59 \pm 0.25$ | $95.71 \pm 0.52$ |
| | Round V.0 Decommitting | 0.70 | 0.67 | 0.77 |
| LightVeriFL | Round V.1 Batch Checking | 0.60 | 5.94 | 61.41 |
| | Verification Phase - Total | $1.30 \pm 0.02$ | $6.61 \pm 0.06$ | $62.18 \pm 0.49$ |
| Gain | | $23.34\times$ | $5.54\times$ | $1.54\times$ |

In Table 4, we investigate the affect of the model size on the proposed verification algorithm. Considering practical FL systems, we take $d = 10K$, $d = 100K$, and $d = 1M$ for $N = 200$ users, 30% dropout rate, and $L = 10$. In this case, decommitting time stays (almost) the same for both of the schemes across all three $d$ values since hashes are of constant length independent of $d$ in either of the schemes. In both of the schemes, the batch checking time, which requires the computation of the hash of the aggregate model, increases linearly with $d$. In Table 4, we see that for smaller $d$ values the verification time is dominated by the decommitting step in `VeriFL`. In these cases, the proposed `LightVeriFL` scheme achieves the biggest gain. In fact, we observe around $23.34\times$ gain when $d = 10K$.

From Tables 1 and 4 we deduce that the gain achieved by the proposed `LightVeriFL` scheme over the `VeriFL` scheme is more prominent when the verification time is dominated by the decommitting step. This happens when the model size is smaller for moderate number of users, e.g., $d = 10K$ and $N = 200$ or $d = 100K$ and $N = 200$. Such a large gain is expected when the number of users gets larger in more complex models with larger model sizes.

