# OpenReview forum: "LightVeriFL: Lightweight and Verifiable Secure Federated Learning"
_NeurIPS.cc/2022/Workshop/Federated_Learning — FL-NeurIPS 2022 Oral_

### Official Review · Reviewer_KjX5 · 2022-10-17
**Extends SecAgg family of protocols to support clients verifying correct summation by server**

This theoretical paper extends the SecAgg family of protocols with a verifiability feature: clients can verify that the sum released by the server is the correctly computed sum of the clients' submitted values.  This prevents an adversarial server from completing the SecAgg protocol, but lying about the results during the release process.   Other properties of SecAgg family algorithms are maintained, such as robustness to users dropping out of the protocol.

The novelty of the paper comes in the efficiency achieved while adding verifiability; existing work has supported the same feature, but with significant communication/computation cost.  In contrast, the algorithm in this paper supports verifiability while being almost as efficient as the underlying SecAgg protocol itself.

This paper makes a strong theoretical advancement, backed by clear presentation, careful analysis, and strong empirical evidence (including demonstrating viability with reasonable scaling parameters).  The opportunity for impact is high, as an important capability is offered at an efficiency nearly matching algorithms that are already in significant use.  Together, this makes an easy case for a strong accept.

---

### Official Review · Reviewer_59Fd · 2022-10-17
**Writing is not very clear, approach seems to have merits for others to consider and to improve on.**

Paper proposes a new verification mechanism for secure aggregation in FL based on HE.
The paper does not describe the setting very clearly, reader needs to refer to Appendix to understand some of the concepts. Also there seems to be typos in formulas here and there. A good paper needs to be succinct in the main manuscript. Appendix is for proofs and tangental topics. Experimental results are also not clearly explained. I have some major concerns about the set-up specifically how FL is described and simulated. Please read my comments below in the Detailed Comments section.

Overall though, the paper has introduced an interesting approach that seems to have merits for others to study and further improve. For this reason I am giving a marginal accept to this submission.

Detailed Comments
---------------------
- From Algorithm (1) it seems that the assumption is that all clients participate in all rounds (module the fact that they can drop out). This is not the proper setting for X-device FL. In X-device client usually there are N >> 1 clients and M << N are randomly selected to participate in a round. The drop-outs can happen in the cohort of M selected clients. If the paper is targeting x-silo FL which is more aligned with the proposal, it should be stated clearly.

- Line 96: What is m in the formula? did you mean h?

- In eq. (1),  are we only using d elements of X, how does this work at all? Text is not clear

- Section 3.1: The definition of FL is very limited specifically with regards to the weights of the updates. L(x) = sum_i a_i L_i(x) is the correct commonly used expression.

---

### Official Review · Reviewer_nQug · 2022-10-18
**Interesting paper, but some details need to be added**

The paper considers federated learning (FL), wherein the server can be malicious and can alter the aggregation process. The paper proposes verifiable and secure aggregation scheme. The key ideas build on [12] and [8], and leverage linearly homomorphic hashing, blinding commitments, and encoded masks.

Strengths:
1. Recent works have shown several attacks that a malicious server can launch in FL, which makes it important to develop verifiable aggregation schemes.
2. The paper is generally well-written.

Weaknesses:
1. The system model and threat model have some missing details. In the proposed scheme, each user share encoded masks and hash commitments with the other users. Does this communication happen through the server? (Typically, in FL, all communication happens through the server.) If yes, then can the server act maliciously when relaying the encoded masks and hash commitments?

2. For computing hashes in eq. (1), model update entries need to be converted to group elements. The paper does not discuss how real numbers will be mapped to group elements, and what can be potential impact on model performance in terms of accuracy.


Other comments:

1. It would be helpful to cite references for attacks that consider a malicious server.

2. Theorem 2 states that LightVeriFL provides input privacy, and then Theorem 4 says that "when LightVeriFL is implemented together with a secure aggregation scheme, a secure verifiable aggregation scheme is obtained". This is quite confusing. Does the proposed scheme need to be combined with an existing secure aggregation to achieve security (i.e., privacy)?

3. Sec. 3.2 first mentions that “All users and the server are honest but curious”. Then, it is mentioned that the server is allowed be forge the aggregation results. This is a bit confusing, and it would be helpful mention the threat model clearly.

4. Before equation (1), $\mathbf{x}$ is considered as a gradient vector, whereas later it is model update. Also, collusion resistance -> collision resistance?

5. If group elements $g_1, g_2, \dots, g_d$ are fixed, then collision probability is computed over which randomness?

---

### Decision · Program_Chairs · 2022-10-20

Accept (Oral)